# A Monoclonal Antibody against the C-Terminal Domain of *Bacillus cereus* Hemolysin II Inhibits HlyII Cytolytic Activity

**DOI:** 10.3390/toxins12120806

**Published:** 2020-12-19

**Authors:** Natalia Rudenko, Alexey Nagel, Anna Zamyatina, Anna Karatovskaya, Vadim Salyamov, Zhanna Andreeva-Kovalevskaya, Alexander Siunov, Alexander Kolesnikov, Anna Shepelyakovskaya, Khanafiy Boziev, Bogdan Melnik, Fedor Brovko, Alexander Solonin

**Affiliations:** 1Pushchino Branch, Shemyakin–Ovchinnikov Institute of Bioorganic Chemistry, Russian Academy of Sciences, 6 Prospekt Nauki, 142290 Pushchino, Moscow Region, Russia; anna.zamjatina@yandex.ru (A.Z.); annakaratovskaya@mail.ru (A.K.); shepelyakovskaya@rambler.ru (A.S.); Bozievhanafiyy@rambler.ru (K.B.); brovko@bibch.ru (F.B.); 2FSBIS FRC Pushchino Scientific Centre of Biological Research, G.K. Skryabin Institute of Biochemistry and Physiology of Microorganisms, Russian Academy of Sciences, 5 Prospekt Nauki, 142290 Pushchino, Moscow Region, Russia; anagell@mail.ru (A.N.); v.salyamoff@yandex.ru (V.S.); hemolysin6@gmail.com (Z.A.-K.); av_siunov@rambler.ru (A.S.); kaczzz@yandex.ru (A.K.); solonin.a.s@yandex.ru (A.S.); 3Pushchino State Institute of Natural Sciences, 3 Prospekt Nauki, 142290 Pushchino, Moscow Region, Russia; 4Protein Institute of the Russian Academy of Sciences, 4 Prospekt Nauki, 142290 Pushchino, Moscow Region, Russia; bmelnik@phys.protres.ru

**Keywords:** pore-forming toxin, hybridoma, bacteriophage display, epitope mapping, neutralizing monoclonal antibody, oligomerization, ELISA, hemolysis, in vivo efficiency

## Abstract

*Bacillus cereus* is the fourth most common cause of foodborne illnesses that produces a variety of pore-forming proteins as the main pathogenic factors. *B. cereus* hemolysin II (HlyII), belonging to pore-forming β-barrel toxins, has a C-terminal extension of 94 amino acid residues designated as HlyIICTD. An analysis of a panel of monoclonal antibodies to the recombinant HlyIICTD protein revealed the ability of the antibody HlyIIC-20 to inhibit HlyII hemolysis. A conformational epitope recognized by HlyIIC-20 was found. by the method of peptide phage display and found that it is localized in the N-terminal part of HlyIICTD. The HlyIIC-20 interacted with a monomeric form of HlyII, thus suppressing maturation of the HlyII toxin. Protection efficiencies of various *B. cereus* strains against HlyII were different and depended on the epitope amino acid composition, as well as, insignificantly, on downstream amino acids. Substitution of L324P and P324L in the hemolysins ATCC14579^T^ and B771, respectively, determined the role of leucine localized to the epitope in suppressing the hemolysis by the antibody. Pre-incubation of HlyIIC-20 with HlyII prevented the death of mice up to an equimolar ratio. A strategy of detecting and neutralizing the toxic activity of HlyII could provide a tool for monitoring and reducing *B. cereus* pathogenicity.

## 1. Introduction

*Bacillus cereus* is the fourth most common cause of foodborne illnesses [1]. Modern phylogeny singles out nine species in seven phylogenetic clades in the *B. cereus* group [2]. *B. cereus* species are associated with outbreaks of foodborne illnesses (*B. cereus* sensu stricto and *Bacillus cytotoxicus*) [3], food spoilage (psychrotolerant *Bacillus weihenstephanensis* and *Bacillus mycoides*) [4] and anthrax disease in humans and animals (*Bacillus anthracis*) [5,6].

*B. cereus* hemolysin II (HlyII) belongs to pore-forming β-barrel toxins [7,8,9,10]. The hemolysin II gene is found in all clades of *B. cereus* sensu lato [11], but is more common among various natural isolates of *Bacillus thuringiensis* [12]. This places HlyII in the focus of the study, since *B. thuringiensis* is widely used as a biological insecticide for plant protection [6]. The toxin forms weakly anion selective pores, whose radius varies from 0.6 to 0.8 nm in both natural erythrocyte membranes and model membranes [13]. In its mature state, this toxin is the closest-known homolog of *Staphylococcus aureus* α-toxin with a 38% identity at the amino acid level [1]. The HlyII protein possesses a C-terminal extension of 94 amino acid residues designated as HlyIICTD (C-terminal domain) [7,14] previously undescribed for this class of toxins. HlyIIΔC shows an 8-fold lower hemolytic activity in affecting rabbit erythrocytes [14]; monoclonal antibodies (mAbs) against HlyIICTD proved useful in revealing the CTD role at various stages of pore formation.

The formation of mAbs against pore-forming proteins is hampered by their toxicity of these proteins to animals during their immunizatin. Previously, these mAbs were produced using inactivated antigens. In this approach, however, the generated mAbs recognized proteins of an artificial structure. Some parts of such proteins can also be used as antigens. In this case, most mAbs obtained also recognize proteins with a structure different from the natural one. According to an assumption in [15], the presence of the C-terminal domain located on the surface of the mushroom-shaped part allows this part of the protein to be used as an antigen. mAbs against toxin fragments, instead of full-length proteins, are potential candidates for developing vaccine antigens [16]. In [17], we presented a panel of mAbs against this part of HlyII. Among these, the antibody HlyIIC-20 was found in the present work to be able to suppress the hemolysis of rabbit erythrocytes by HlyII. Here, we describe this mAb against HlyIICTD and determine its characteristics.

## 2. Results

### 2.1. Characteristic Features of the Monoclonal Antibody HlyIIC-20

The work [17] describes the formation of a panel of mAbs against the C-terminal domain of *B. cereus* HlyII, using to which the HlyIICTD is shown to be capable of independently binding erythrocyte membranes.

All 24 antibodies from the formed panel have been tested for the ability to influence the hemolytic activity of HlyII. The mAb HlyIIC-20, which contains a heavy chain γ2b and a light chain κ, inhibits hemolytic activity (GenBank accession numbers MW194175 and MW194176).

The antibody HlyIIC-20 recognizes HlyIICTD, both the native form of HlyII and the denatured forms of these proteins, in immunoblotting assays. The affinity constants characterizing the interaction of this antibody with HlyIICTD, intact HlyII14579 and HlyII771 were determined by indirect ELISA as described in [18]. Before the measurements and all other antibody experiments, the samples were checked for aggregates by dynamic light scattering. The analysis of the particle size distribution showed that the solution contained mainly particles corresponding to the molecular weight of the antibodies, without a significant number of high-molecular aggregates. Figure 1 shows a plot for determining the affinity constant for the interaction of HlyIIC-20 with HlyII14579. The affinity constant characterizing the interaction of HlyIICTD with HlyIIC-20 was 3.4 × 10^9^ M^−1^; with HlyII14579, 2.6 × 10^9^ M^−1^; with HlyII771, 1.8 × 10^9^ M^−1^.

Based on the determined coefficients, it can be concluded that the degree of accessibility of the antigen epitope for natural hemolysins varies, and it differs from the recombinant HlyIICTD. The efficiency of recognition of the ATCC14579^T^ and B771 hemolysins II by the antibody HlyIIC-20 differs by 30%, while the binding efficiency of the antibody to the recombinant HlyIICTD of ATCC14579^T^ hemolysin is approximately 25% higher than that for the full-length protein.

An excess of HlyIIC-20 did not inhibit the interaction of biotinylated HlyIICTD with erythrocytes, which indicates that the antigenic determinant on the surface of HlyIICTD recognized by this antibody is located at a site different from the site of target cell binding.

### 2.2. Dependence of HlyII Oligomerization on the Presence of HlyIIC-20

In the absence of erythrocyte membranes in solution, full-length HlyII is monomeric (Figure 2, lane 1). HlyII is characterized by the formation of stable oligomeric forms (Figure 2, lane 2) upon interaction with the target cell membrane [14]. Pretreatment of the full-length toxin with HlyIIC-20 prevents the formation of stable oligomeric forms (Figure 2, lane 3) during its interaction with the erythrocyte membrane, which means that the HlyIIC-20 epitope is located in the region of the molecule important for oligomerization of the monomeric form of the toxin into a full pore on the cell membrane and, consequently, the formation of a channel.

### 2.3. Identification of the Primary Sequence-Dependent Epitope Recognized by HlyIIC-20

Phage display revealed six sequences that could be epitopes recognized by HlyIIC-20 (Figure 3). Analysis of the sequences in Figure 4 revealed no identical motifs, although all of them interacted with HlyIIC-20. This suggests that these epitopes are conformational, i.e., in the sequences it is necessary to look for a similar composition of amino acids, and their correct mutual arrangement occurs only when the three-dimensional (3D) structure is formed. When analyzing the composition of these sequences, we assumed that the formation of an epitope required a pair of amino acid residues with amino groups (N, Q or H) and a positively charged amino acid lysine or arginine (K, R). Figure 4 shows a conformational model of the C-domain (PDB: 6D5Z), in which amino acids N, Q and K are highlighted (see the legend to Figure 5). It should be noted that there are no histidines (H) in the amino acid sequence of the C-domain. An analysis of amino acid localization showed that the protein has only two places on the surface, where all the three amino acids, N, Q and K, are close together. One is at the beginning of HlyIICTD and the other in its middle part (see Figure 5). Phage display revealed two putative epitopes: NQKALEEQ and NGNQLK (Figure 4). Figure 5 demonstrates a conformational model of the C-domain of HlyII with two regions where the amino acid residues N, Q and K are close to one another. Figure 5B shows a 3D structure containing the amino acids that can presumably form an epitope at the beginning of the C-terminal domain; these are amino acids N320–Q327 and, in contact with them, N377 and Y396. Figure 5D shows a 3D structure containing the amino acids that can presumably form an epitope in the middle of the C-domain; these are amino acids N350–K355 and N401, L403 in contact with them. The same areas are underlined by black lines in Figure 4. To check which of these two putative epitopes is functional, this region of HlyII was compared with each epitope in three natural strains B771, ATCC14579^T^ and ATCC4342^T^. The latter putative epitope was found to be identical in hemolysins of all three strains, while the former had one amino acid substitution. Hemolysin II of ATCC 14579^T^ and ATCC 4342^T^ contained an identical amino acid composition in this region, and in B771 HlyII, the amino acid residue of leucine was replaced by proline. The region located between these putative epitopes contains variable amino acid residues that are almost identical in B771 and ATCC4342^T^ hemolysins. To check which of the putative epitopes is true, reciprocal mutations L324P and P324L were introduced into ATCC14579^T^ and B771 hemolysins, respectively (Figure 4).

### 2.4. Inhibition of HlyII in Experiments In Vitro

In used natural strains, the putative conformational epitope has several amino acid substitutions both within and away from its linear region (Figure 4). A set of constructs were made to verify the theoretically solved structure of the putative conformational epitope. The constructed ATCC14579^T^ and B771 hemolysin mutants were tested for the level of protection of rabbit erythrocytes against hemolysis by HlyIIC-20. Comparison of the curves in Figure 6A,C shows that the level of protection against hemolysis by the mutant form of ATCC14579^T^ hemolysin II is significantly reduced. Thus, the presence of the amino acid L324 is essential for the suppression of hemolysis. The affinity coefficient of HlyIIC-20 with B771 HlyII differs slightly from that for ATCC14579^T^. Introduction of the substitution P324L into B771 hemolysin II confirmed the role of leucine in epitope recognition, since the antibody suppressed hemolysis in mutant more efficiently than in the wild type. The presence of proline in the linear part of the epitope impairs recognition by 30%, but the inhibition level of hemolysis significantly decreases (up to 10 times). The role of leucine in the linear part of the conformational epitope in protection against hemolysis in the presence of HlyIIC-20 was confirmed by using hemolysin from the ATCC4342^T^ strain (Figure 4 and Figure 6E). The influence of the proline amino acid residue (P324) on the recognition of the epitope by HlyIIC-20 and its effect on the decrease in the level of protection by the antibody is apparently determined by the possible existence of this residue in two isoforms. The *cis* and *trans* isoforms of proline are able to alter the 3D structure of proteins that contain proline residues [19]. The ratio of the isoforms is determined by the state of the peptidyl-proline-*cis*-*trans* isomerase of the bacterial cell [20] and can additionally regulate the functional activity of bacterial proteins by changing their structure. HlyIIC-20 recognizes a conformational epitope located at the N-terminus of HlyIICTD. The presence of proline in its content in some natural hemolysins reduces both the efficiency of antibody recognition and the level of protection against hemolysis due to its effect on the protein conformation.

The treatment of rabbit erythrocytes with hemolysin on ice followed by washing with cold PBS showed that HlyIIC-20 could not protect erythrocytes from hemolysis. At room temperature, the antibody was shown to inhibit the hemolytic activity. The reason is, evidently, that the described antibody binds to HlyII in a monomeric form both in the solution and on the membrane.

### 2.5. Inhibition of HlyII in Experiments with Mice

Experiments on the inhibition of the toxic activity of HlyII (ATCC14579^T^) in vivo were carried out using white BALB/c mice. The animals were injected intravenously through the tail vein with a dose of toxin equal to LD_50_ and the corresponding amount of toxin pre-incubated with excess HlyIIC-20. In the case of using HlyII (ATCC14579^T^), after 24 h observation, all mice that received an injection of the toxin incubated with all dilutions of antibodies, including that with an equimolar number of antibodies, remained alive. In the control group, 50% of animals injected with a toxin dose of LD_50_ were alive, while the deviation in each individual experimental group was no more than 25% with a hemolysin dose of 200 U per animal at an intravenous injection of purified ATCC14579^T^ HlyII. This result evidenced the toxin-neutralizing activity of HlyIIC-20 for HlyII (ATCC14579^T^). When the HlyII toxin from *B. cereus* B771 was used for injection, no noticeable neutralizing activity of HlyIIC-20 was observed. This survival rate suggests the possibility of developing antidotes against HlyII based on this antibody.

As seen in Figure 6A,B, the efficiency of rabbit erythrocyte protection against hemolysis upon addition of HlyIIC-20 for hemolysins from strains ATCC14579^T^ and B771 differs significantly. Under conditions of the experiment, rabbit erythrocytes are protected by HlyIIC-20 from 10 U ATCC14579^T^ HlyII, whereas for B771 HlyII, in the presence of HlyIIC-20, they are lysed immediately upon addition of less than 1 unit of HlyII. Thus, the results obtained on experimental animals agree with those described in the previous sections.

## 3. Discussion

Pore-forming toxins are the most important virulence bacterial factors They represent an attractive target for the development of molecules that neutralize their actions with high efficacy [21,22]. Monoclonal antibodies are promising as highly specific and reproducible tools for neutralizing virulent factors [23]. The mAbs can be broadly cross-reactive and recognize several pore-forming toxins [24]. Despite the significant number of described neutralizing monoclonal antibodies against bacterial toxins [25,26], in the fight against bacterial infections, these antibodies are not yet widespread, in contrast to other fields, such as oncology and viral infections. Research activities towards developing novel strategies for the diagnosis and suppression of pore-forming toxins allow developing the anti-virulence therapy. Unlike in other studies, in our work, a separate fragment of the toxin molecule was used to obtain antibodies, which has been shown to be of importance for the process of hemolysis. A strategy of detecting and neutralizing the toxic activity of HlyII could provide a potent tool for monitoring and reducing *B. cereus* pathogenicity [27]. The first involves the detection of *B. cereus* HlyII in food and biological fluids [28]. The latter supposes the development of approaches to anti-virulence therapy, which requires the study of all stages of pore formation: regulation of the expression of genes encoding toxin, secretion of monomeric forms from a bacterial cell into the external environment, as well as stages of pore formation during an attack by eukaryotic cells.

This paper describes HlyIIC-20 from the mAb panel against HlyIICTD that recognizes a conformational epitope and is able to protect rabbit erythrocytes against hemolysis. Oligomerization of the HlyII toxin is a necessary stage in the formation of pores and determines the possibility of hemolysis of rabbit erythrocytes [14]. Based on the analysis of the effectiveness of erythrocyte protection against hemolysis by hemolysins of various *B. cereus* strains, it can be concluded that the efficiency of hemolysis in the presence of HlyIIC-20 depends on the primary structure of the region of conformational epitope recognized by this antibody. This work presents a comparative analysis of these regions in hemolysins from *B. cereus* ATCC14579^T^, B771 and ATCC4342^T^ strains. The amino acid composition downstream of the linear region of the epitope varies in different hemolysins. In this region, the compositions of the ATCC4342 and B771 hemolysin II were almost identical and different from that of ATCC14579^T^ hemolysin II. However, for HlyII from the strains ATCC14579^T^ and ATCC4342^T^, the level of hemolysis inhibition was relatively the same, while the sensitivity of HlyII from B771 to the presence of HlyIIC-20 was noticeably lower. The linear part of the epitope in B771 hemolysin II has a natural replacement of Leu324 by Pro, which apparently determines the revealed differences in the level of recognition by HlyIIC-20 and protection against hemolysis. Introduction of the reciprocal mutations L324P and P324L into ATCC14579^T^ and B771 hemolysins, respectively, changed their sensitivity to HlyIIC-20. The sensitivity of the mutant form of ATCC14579^T^ HlyII to the presence of the antibody decreased, while that of the B771 mutant increased significantly. The proline amino acid residue usually exists in proteins in two isoforms. One amino acid substitution in the epitope region reduced the recognition efficiency and inhibition of hemolysis in the presence of HlyIIC-20. The formation of mAbs based on a part of the toxin protein allows further use of these antibodies to identify toxins and to neutralize their action.

## 4. Conclusions

An antibody capable of forming an immune complex in aqueous solution was found in a panel of monoclonal antibodies to the *B. cereus* recombinant HlyIICTD protein. HlyII pretreated with the antibody HlyIIC-20 inhibits the formation of oligomers and decreases the cytolytic activity of HlyII. The level of inhibition depends on the origin of HlyII. A comparison of HlyIICTD from various *B. cereus* strains revealed the presence of a variable region in the conformational epitope. The introduction of the reciprocal mutations L324P and P324L into this region of hemolysin II from the *B. cereus* ATCC14579^T^ and B771 strains, respectively, suggested the importance of these amino acid residues for pore maturation. Since the binding efficiency of HlyIIC-20 with respect to various hemolysins and HlyIICTD was different, and the antibody-caused suppression of hemolysis also varied, we assume a significant role of leucine located within the epitope in HlyII pore maturation at the step of oligomerization. The obtained data confirm the structure of the conformational epitope. HlyIIC-20 suppresses the HlyII-induced deaths of mice in quantities up to the equimolar pre-incubated HlyII/antibody ratio.

## 5. Materials and Methods

### 5.1. Plasmid Strains and Proteins

Bacillus cereus ATCC 14579^T^, Bacillus cereus ATCC 4342^T^, Bacillus cereus B771 [29].

*E. coli* BL21 (DE3) (Novagen, Germany) was used to transform pET29b (+) (Novagen, Darmstadt, Germany), *E. coli* ER2738 for affinity phage selection.

### 5.2. Media and Solutions

Medium: 2YT (16 g/L bactotryptone, 1 g/L yeast extract, 5 g/L NaCl, pH 7.0), 1.5% agar and 0.7% agar based on 2YT. Solutions: 1000 × IPTG/X-gal (1.25 g IPTG (isopropyl-β-d-thiogalactoside) and 1 g X-gal (5-bromo-4-chloro-3-indolyl-β-d-galactoside, Sigma, USA) in 25 mL DMF (dimethyl formamide, Sigma, USA), tetracycline 20 mg/mL (Sigma, USA) in 50% ethanol, phosphate buffered saline (PBS), PBST—PBS containing 0.1% Tween-20 (Sigma, USA), blocking solution of 1% gelatin (Sigma, USA) based on PBST, PEG/NaCl precipitating solution (20% (w/v) polyethylene glycol-6000, Sigma, USA, 2.5 M NaCl). Protein markers (Abcam, GB) and DNA electrophoresis markers (Fermentas, Lithuania), conjugate of streptavidin with horseradish peroxidase (Thermo Fisher Scientific, USA).

### 5.3. Determination of the Primary Sequence of CDR (Complementarity-Determining Region) Antibodies

Hybridoma cells (10^6^) producing antibodies that inhibit ATCC14579^T^ HlyII hemolysis were selected. The hydridoma cells were washed twice with PBS, supplemented with 1 mL of TRIzol (Invitrogen, Carlsbad, CA, USA) and frozen at –70 °C. RNA was isolated by the phenol-chloroform method using TRIzol reagent (Thermo Fisher Scientific, USA) according to the manufacturer’s protocol (Thermo Fisher Scientific, USA) [30,31]. The purity and amount of RNA was assessed spectrophotometrically. Reverse transcription was performed according to [32] using SuperScript III reverse transcriptase (Invitrogen, USA) for 90 min at 42 °C. Next, the touchdown PCR with Q5 polymerase (NEB, USA), GC buffer and a proprietary set of primers (Biogen, Cambridge, MA, USA) was used. The heavy chain was sequenced using PCR primers at Evrogen (Russia). Due to the doubling of light chain sequences, the PCR product was excised from the gel and then integrated into SmaI site of pUC18 vector after blunt ends dephosphorylation using T4 ligase (NEB, Ipswich, MA, USA). The reaction was stopped by heating the mixture to 65 °C, and the mixture was used to transform chemically competent cells *E. coli* XL1-blue (Evrogen, Moscow, Russia). Selected clones were sequenced by Evrogen.

### 5.4. Production and Isolation of mAbs against HlyIICTD

MAbs were isolated by affinity chromatography on protein A sepharose (Thermo Fisher Scientific, USA) [33] from hybridoma culture fluids secreting the mAbs. Then, the samples were centrifuged at 12,000× *g* at +4 °C. Dynamic light scattering was measured by a particle size analyzer from “Malvern Zetasizer Nano ZSP.” The thermostated cuvette had a size of 0.3 × 0.3 cm and a volume of 100 μL. Light scattering was measured at an angle of 173 degrees. The data were analyzed with Malvern software.

The types of heavy and light chains of immunoglobulins were determined by ELISA (Termo Fisher Scientific, USA) according to the manufacturer’s instructions.

### 5.5. Conjugation of Antibodies with Biotin

HlyIIC-20 and HlyIICTD were biotinylated using a solution of biotin N-hydroxysuccinimide ester (Sigma, St. Louis, MO, USA) in dimethyl sulfoxide at a concentration of 2.9 mM. Biotin ether was added at a 20-fold molar excess in relation to antibodies. The mixture was incubated for 4 h at room temperature. To remove the unreacted reagent, the mixture was dialyzed against PBS overnight.

### 5.6. Immunoblotting

HlyII toxin (0.3 µM) was incubated with HlyIIC-20 (1 µM) for 1 h at 37 °C, then erythrocytes were added to a final concentration of 0.5% and incubated for 1 h at 37 °C with stirring. Control samples contained no HlyIIC-20. Electrophoretic separation of proteins was carried out in the presence of β-mercaptoethanol (Sigma, USA) as in [34]. Transfer to a nitrocellulose membrane was carried out for 15 h at a current of 20 mA in a buffer containing 25 mM Tris–HCl, 0.25 M glycine, 0.1% sodium dodecyl sulfate, 20% methanol, pH 8.3. Centers of nonspecific sorption were blocked by adding 1% (*w*/*v*) gelatin solution in PBST for 30 min. Then, the membrane was incubated for 2 h with biotinylated antibodies HlyIIC-20 (10 μg/mL). After incubation, the membrane was treated for 1 h with streptavidin conjugated with horseradish peroxidase diluted in PBST according to the manufacturer’s recommendation (Thermo Fisher Scientific, Waltham, MA, USA). At each stage, the membrane was thoroughly washed with PBST. The membrane was stained with a solution containing 3 mM diaminobenzidine-3,3 tetrahydrochloride (Sigma-Aldrich, St. Louis, MO, USA) and 0.03% hydrogen peroxide.

### 5.7. Peptide Phage Display

Peculiarities of the interaction of HlyIICTD with erythrocytes revealed with the use of HlyIIC-20 set the task of determining the epitope recognized by this antibody. The antigenic determinant HlyIICTD recognized by HlyIIC-20 was determined by the peptide phage display method as described in [35]. We used a library of random peptides of 12 amino acid residues (NEB, Ipswich, MA, USA) displayed on the M13KE phage [36]. The 12-dimensional phage peptide library has a repertoire of 109; peptides are exposed on the surface of bacteriophages in the composition of the minor protein pIII. Each clone in the library is represented by ~1000 copies. Selection of phages carrying peptide amino acid sequences specifically interacting with HlyIIC-20 was performed according to their ability to interact with one another. Three rounds of selection were carried out. The DNAs of the selected individual phage clones were sequenced. Using the GeneRunner program, the obtained nucleotide sequences were analyzed and the amino acid sequences of the peptides recognized by HlyIIC-20 and exposed in the pIII protein of bacteriophages were determined. After analyzing their interaction with the mAb, 10 clones were selected, in which the sequences of the peptide displayed by the phage were determined. Affinity selection of bacteriophages, cultivation of bacteriophages in semiliquid agar and isolation of bacteriophage DNA were performed as in [37].

### 5.8. DNA Sequencing

Analysis of the DNA sequences of the descendants of phage clones to determine the sequence of the insertions was carried out at the Evrogen (Russia). Sequencing results were processed using the Gene Runner 6.5.52 and ClustalW programs.

### 5.9. Interaction of Biotinylated HlyIICTD with Erythrocytes in the Presence of HlyIIC-20

The effect of the antibody HlyIIC-20 on the interaction of HlyIICTD with erythrocytes was studied in the reaction carried out in PBS containing 10% bovine serum as follows: 1.7 μM HlyIIC-20 was preincubated with 0.39 μM HlyIICTD-bio for 1 h at 37 °C, then a suspension of erythrocytes was added to a concentration of 0.025%, and the mixture was incubated for 1 h at 37 °C. Washing was carried out with 5% bovine serum albumin solution in PBS to remove a possible nonspecific interaction. A total of 100 μL of the solution was added to the wells and centrifuged for 10 min at 3000 rpm. The reaction was developed with streptavidin conjugated with horseradish peroxidase.

### 5.10. HlyII Neutralization Assay

The hemolytic activities of wild type HlyII from various *B. cereus* strains (ATCC 14579^T^, ATCC4342^T^ and B771) and mutant forms of recombinant hemolysin II (ATCC14579^T^ L324P, B771 and P324L) and their neutralization by HlyIIC-20 were investigated using rabbit erythrocytes [38]. Lysates of induced bacterial cells carrying recombinant plasmids with natural and mutant HlyII at a stepwise twofold dilution in a volume of 45 μL were incubated for 20 min at 37 °C in the presence of 5 μL of 6.7 μM HlyIIC-20 or PBS, then added to 50 μL of 1% rabbit erythrocytes in PBS. Analysis of the neutralization of HlyII with the antibody HlyIIC-20 after the integration of HlyII monomers into the erythrocyte membrane was performed according to the following scheme. After binding 0.5% suspension of erythrocytes to HlyII on ice, the suspension was divided into two parts and sedimented. One part was resuspended in PBS with HlyIIC-20 to a final concentration of 0.33 μM, the other part—in the antibody-free buffer. Both parts were incubated at 37 °C for 30 min. The hemolytic activity was measured as described in [38].

The 0.5% suspension of erythrocytes was bound to HlyII at room temperature in two test tubes and sedimented; one pellet was suspended in PBS with HlyIIC-20 to a final concentration of 0.67 μM, the other was suspended in PBS. The reaction mixture was kept for 5 min at room temperature and incubated at 37 °C for 30 min. Then, the hemolytic activity was measured [38].

### 5.11. Animal Experiments

The in vivo toxin neutralization assay was performed using female 6–8-week-old BALB/c mice about 20 g each, five experimental groups with four mice per group. The animals were used according to the protocol: “Search for toxin-neutralizing antibodies to bacterial toxins in a mouse model,” registration number 674/18 of 08.10.2018, approved at a meeting of the Commission for Control over the Maintenance and Use of Laboratory Animals of the BIBCh RAS on 25 December 2018. These studies were planned for two years. Animals were obtained from the Laboratory Animal Breeding Nursery, Pushchino Branch, Institute of Bioorganic Chemistry, Russian Academy of Sciences, which has earned the international AAALACi accreditation. Toxins were applied in 100 μL PBS for intravenous challenge. Minimal lethal doses were determined in experiments using serial dilutions of toxins. For BALB/c mice, LD_50_ was experimentally picked (the half-lethal dose is the average dose of a substance that causes the death of half of the members of the test group), which was 200 hemolytic units in 100 μL saline solution per animal. In an experiment for the in vivo toxin neutralization assay, groups of BALB/c mice were passively injected intravenously into the tail vein with serial twofold dilutions of HlyII. This amount of toxin was incubated with a 20-, 10-, 5- and 2-molar excess and the equimolar amount of HlyIIC-20 for 40 min at room temperature. Then, mixtures of antibodies with toxin, as well as toxin at a dose of LD_50_, were injected. The mice were observed up to 24 h and all deaths were recorded, during which time the animals were kept in their usual conditions of detention with free access to water and food. 

## Figures and Tables

**Figure 1 toxins-12-00806-f001:**
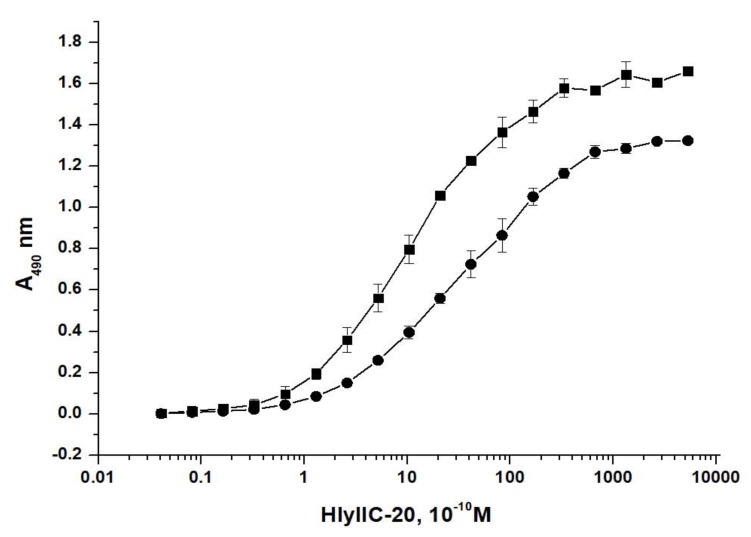
A plot of titration of the antibody HlyIIC-20 for determining its affinity constant by the method of [18] with 10 (●) and 20 (■) ng/well HlyII ATCC14579^T^.

**Figure 2 toxins-12-00806-f002:**
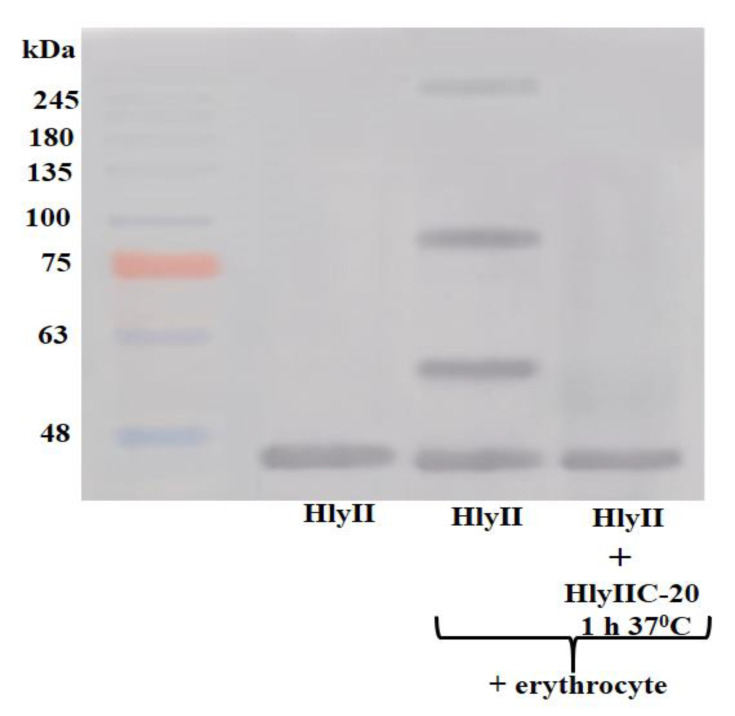
Immunoblotting of HlyII B771 in the presence of erythrocytes with and without preliminary incubation with HlyIIC-20 stained with the biotinylated form of HlyIIC-20 and streptavidin conjugated with horseradish peroxidase.

**Figure 3 toxins-12-00806-f003:**
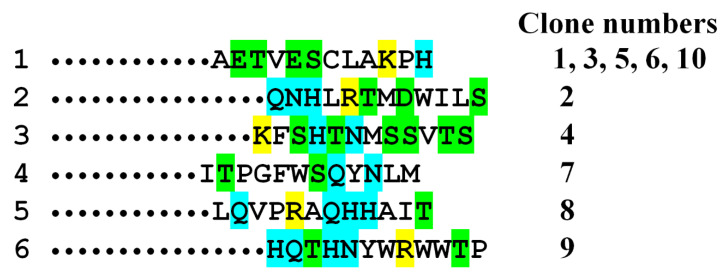
Alignment of the HlyIICTD amino acid sequence region from HlyII *Bacillus cereus* ATCC14579^T^ and sequenced peptides. Charged amino acids are highlighted in yellow, amino acids containing hydroxyl groups are green and amino acids containing amino groups are cyan. The amount and number of clones containing an individual peptide are shown on the right.

**Figure 4 toxins-12-00806-f004:**
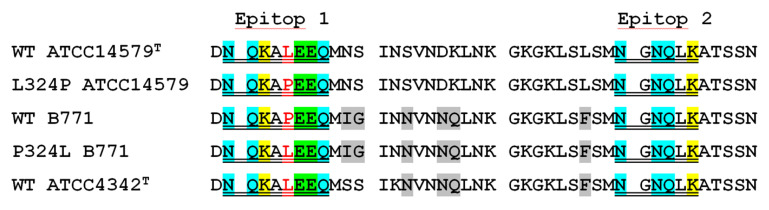
Sequence alignment of *Bacillus cereus* sensu lato HlyIICTD (ATCC14579^T^, B771, ATCC4342^T^) and mutant forms of the latter (numbered from 319 aa to 360 aa according to the full length HlyII toxin [7]). Putative binding sites (linear part of epitops) for the neutralizing HlyIIC-20 are double underlined: aa N320-Q327 and N350-K355. Amino acid residues within these regions are highlighted as in Figure 3. Identical amino acid residues for B771 and ATCC4342^T^ are in grey. Amino acid substitutions for HlyIICTD are marked as red.

**Figure 5 toxins-12-00806-f005:**
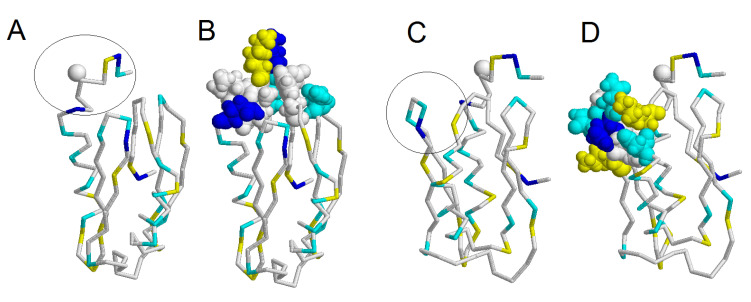
A model of HlyII C-domain (PDB: 6D5Z). Figures (**A**) and (**B**) differ from Figures (**C**) and (**D**) by a rotation of 90 degrees. Amino acids N, Q and K are drawn together, circled; amino acids N, Q and K are highlighted in cyan, blue and yellow, respectively. Amino acid L324 of hemolysin II strain ATCC14579^T^ is shown as a grey ball. Figure **B** shows a three-dimensional (3D) structure containing amino acids that can presumably form an epitope at the beginning of the C-terminal domain; these are amino acids N320–Q327 and, in contact with them, N377 and Y396. Figure **D** shows a 3D structure containing amino acids that can presumably form an epitope in the middle of the C-domain; these are amino acids N350–K355 and, in contact with them, N401 and L403.

**Figure 6 toxins-12-00806-f006:**
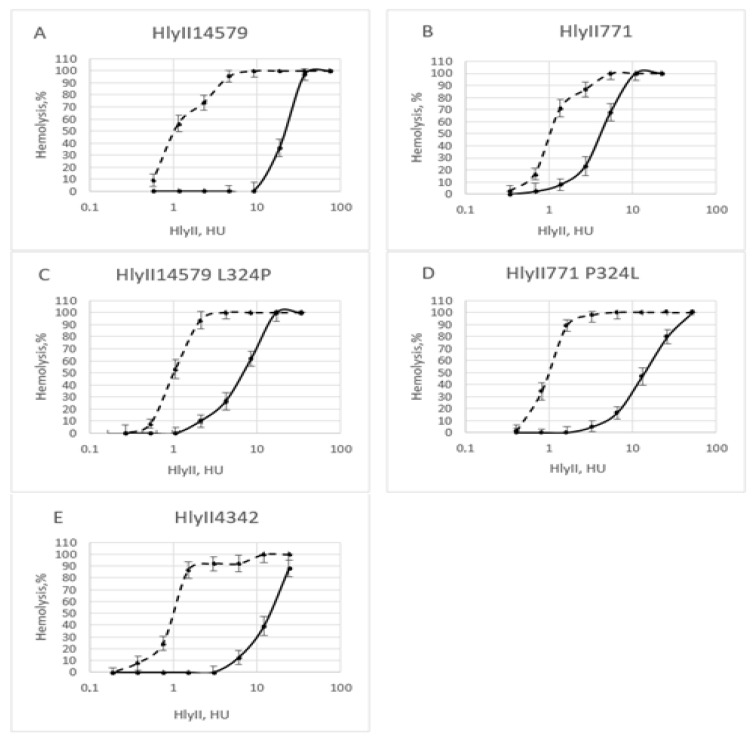
Suppression of hemolysis by HlyIIC-20 during the attack on rabbit erythrocytes by various hemolysins. (**A**) ATCC14579^T^ HlyII before (dotted line) and after (solid line) addition of HlyIIC-20. (**B**) B771 HlyII; (**C**) ATCC14579^T^ with mutation L324P; (**D**) B771 P324L; (**E**) ATCC4342^T^ HlyII. Averaged data from five experiments are presented.

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
