# Peer review of "A Monoclonal Antibody against the C-Terminal Domain of Bacillus cereus Hemolysin II Inhibits HlyII Cytolytic Activity"

_toxins, 2020, doi:10.3390/toxins12120806_

Round 1

Reviewer 1 Report

Dear authors, 

your manuscript is ok for publication in a large part. Although, i have some suggestions to improve its quality, see my comments below. In addition, I insist on a completely rewritten Discussion section. The discussion at the current state does not fulfill the Standard required for a scientific publication.

Figure 2: it should be mentioned what is seen here. Are the multiple bands depicting the oligomerization of Hly?

Line 97: Figure 4 is mentioned. are you sure? do you mean Figure 3 instead?

Lines 97, 99, 101, 103: the repeated mentioning of Figure 4 is redundant

Figure 3: there is no corresponding reference of Figure 3 in the text!

Figure 3: please add in the legend what the  colors yellow, green, blue should mean

Figure 3/4:  please standardize aa residues in either capital- or small letters

Figure 4: please add numeration of the aa residues

Figure 4: the epitopes NQKALEEQ and NGNQLK should be pointed out more distinctly, as they are referred to Line 108

Line 150: no deviation? it was always exactly 50% of the total number of mice?

Discussion: To me, the Discussion is not acceptable. Are there really no references, no previous study you can discuss and set your results in context with? Additionally, its largely a Repetition of the Results. If the article should be published, I insist on a completely rewritten Discussion section including References 

Line 221: weight of the animal?

Conclusion: Again, to me this is no conclusion but again a redundant repetition of what has already been mentioned in Results and Discussion. 

Line 243/244: better intergrate this part into the discussion and explain detailly (also with references!) , what such a strategy could look like

Materials and Methods: Please add the countries to each Company/manufacturer. Please also standardize (United States OR USA).

Line 262: Explain the method at least in brief: how do you lyse cells? do you use a kit or do you use a phenol/chloroform extraction method?

Lines 270-273: too short, describe more detailly. other scientists should be able to reproduce your results by just reading your method section

Lines 275-278: this paragraph is too short, explain the method more detailly

Line 281: concentration of what? molar? weight?

Line 305: explain at least in brief

Line 307-309: please explain the method more detailly

Line 309: versions? companies? countries?

Author Response

The Authors are grateful to Reviewer #1 for detailed analysis and valuable comments and remarks. All Reviewer’s remarks were considered and respective corrections/changes were made.

Figure 2: it should be mentioned what is seen here. Are the multiple bands depicting the oligomerization of Hly?

According to your remarks, we redesigned the Figure 2 and made the appropriate changes to the text in section 2.2, lines 96-103.

Line 97: Figure 4 is mentioned. are you sure? do you mean Figure 3 instead?

Thanks for your comment, we have included the link to Figure 3 in the text (section 2.3, lines 111).

Lines 97, 99, 101, 103: the repeated mentioning of Figure 4 is redundant

Thanks for your comment, we have reduced the excessive number of references to figure 4 in the manuscript.

Figure 3: there is no corresponding reference of Figure 3 in the text!

Thanks for your comment, we have included the link to Figure 3 in the text (section 2.3, lines 111).

Figure 3: please add in the legend what the  colors  yellow, green, blue should mean

Thanks for your comment, we аdded to figure 3 legend (lines 142-143).

Figure 3/4:  please standardize aa residues in either capital- or small letters

Thanks for your comment, we used capital letters in Figures 3 and 4.

Figure 4: please add numeration of the aa residues

            Thanks for your comment, we add numeration of the aa residues in figure 4 legend (line 147).

Figure 4: the epitopes NQKALEEQ and NGNQLK should be pointed out more distinctly, as they are referred to Line 108

Thanks for your comment, the epitopes NQKALEEQ and NGNQLK indicated as epitope 1  and epitope 2, respectively, in the figure 4.

Line 150: no deviation? it was always exactly 50% of the total number of mice?

Thanks for your comment, in the experimental groups of animals, after the injection of the toxin preincubated with the antibody HlyIIC-20, all the mice remained alive. Тhe deviation was observed only in the control group, the deviation in each individual experimental group was no more than 25%, this fact is indicated in the text (section 2.5, lines 214-216).

Discussion: To me, the Discussion is not acceptable. Are there really no references, no previous study you can discuss and set your results in context with? Additionally, its largely a Repetition of the Results. If the article should be published, I insist on a completely rewritten Discussion section including References 

            Discussion rewritten according to the comments of the reviewer

Thanks for your comment, we were considered all your comments, we entirely rewrote the discussion in the final text of the manuscript, removed the repetitions and added reference links for discussion.

Line 221: weight of the animal?

            Thanks for your comment, the weight indicated, about 20g, (section 5.11, line 428).

Conclusion: Again, to me this is no conclusion but again a redundant repetition of what has already been mentioned in Results and Discussion. 

Thanks for your comment, we shortened the сonclusion.

Line 243/244: better intergrate this part into the discussion and explain detailly (also with references!) , what such a strategy could look like

            Thanks for your comment, this sentence: “Formation of monoclonal antibodies based on a part of the toxin protein enables the development of a strategy of using these antibodies to identify toxins and to neutralize their action” have moved, expanded and discussed in the discussion section.

Materials and Methods: Please add the countries to each Company/manufacturer. Please also standardize (United States OR USA).

Thanks for your comment, we've added countries/manufacturers for each company and standardized the country names.

Line 262: Explain the method at least in brief: how do you lyse cells? do you use a kit or do you use a phenol/chloroform extraction method?

            Thanks for your comment, for RNA-isolation hybridoma  cells was lysed by the phenol-chloroform method using TRIzol reagent according to the manufacturer's protocol (Thermo Fisher Scientific, USA) [Rio, D.C.; Ares, M.; Hannon, G.J.; Nilsen, T.W. Purification of RNA using TRIzol (TRI reagent). Cold Spring Harb Protoc 2010, 2010, pdb.prot5439, doi:10.1101/pdb.prot5439. Chomczynski, P.; Sacchi, N. Single-step method of RNA isolation by acid guanidinium thiocyanate-phenol-chloroform extraction. Anal Biochem 1987, 162, 156–159, doi:10.1006/abio.1987.9999.] (section 5.3 lines 324-326).

Lines 270-273: too short, describe more detailly. other scientists should be able to reproduce your results by just reading your method section

            Thanks for your comment, methodology has been expanded (section 5.3).

Lines 275-278: this paragraph is too short, explain the method more detailly

Thanks for your comment, methodology has been expanded (section 5.4, lines 347-351).

Line 281: concentration of what? molar? weight?

Thanks for your comment,  we have indicated the molar concentration for the biotin N-hydroxysuccinimide ester (Section 5.5, line 356 )

Line 305: explain at least in brief

Line 307-309: please explain the method more detailly

Line 309: versions? companies? countries?

Thanks for your comments, these remarks refer to the phage display method. We have expanded the description of this  methodology, however detailed in articles:

Zamyatina, A.V.; Rudenko, N.V.; Karatovskaya, A.P.; Shepelyakovskaya, A.O.; Siunov, A.V.; Andreeva-Kovalevskaya, Zh.I.; Nagel, A.S.; Salyamov, V.I.; Kolesnikov, A.S.; Brovko, F.A.; et al. HlyIIC 15 monoclonal antibody against the C-terminal domain of B. cereus HlyII interacts with the thrombin recognition site. Russian Journal of Bioorganic Chemistry2020, 46, 1214–1220.

       Ph.D.TM Phage Display Libraries. Instruction Manual. Version 3.0 11/18. New England Biolabs, Inc.

Unfortunately, the work of Zamyatina et al., 2020 has not yet been published in the English version of the journal "Russian Journal of Bioorganic Chemistry". We have referred to make it available to future readers of the journal. We have only the author's proofreading of this article in English. We send to the editor of the journal “Toxins” english versions of this articles.

Reviewer 2 Report

Comments Toxins-1002352

The authors investigated the effects of monoclonal antibodies on recombinant hemolysin II (HlyII) produced originally by Bacillus cereus, a bacterium causing foodborne illnesses. One of the antibodies termed HlyIIC-20 recognized a conformational epitope within the C-terminal part of HlyII. This monoclonal antibody was able to block HlyII-mediated hemolysis. The authors found out that the antibody HlyIIC-20 blocked maturation of HlyII (i.e. pore formation). Mutagenesis within the conformational epitope of hemolysin II suggested that leucine in position 324 plays a key role in the block of hemolysis by HlyIIC-20.

  1. The work presented here by the authors is largely based on the work in ref. 17 published in Russ J Bioorg Chem. This publication is not available in PubMed or in any other database, which is accessible to the reviewer. Similarly, ref. 25 (Zamyatina et al., 2020) may contain material of the Ms., which could mean that part of the study described above is already published.
  1. This reviewer found a note in research gate of another paper of the same authors that is not cited here (Russ J Bioorg Chem. 46(3):280-285), but contains important information concerning the interaction of the C-terminal end of HlyII with different monoclonal antibodies and rabbit erythrocytes. In this publication the authors argue that the deletion of the 94 amino acids from the C-terminus blocks pore formation by HlyII.

Author Response

The Authors are grateful to Reviewer #2 for detailed analysis and valuable comments and remarks.

We apologize, the works of Rudenko et al., 2020 and Zamyatina et al., 2020 were published in the Russian version of the journal «Bioorganic Chemistry». The work of Rudenko et al., 2020 has already been published in the English version. The work of Zamyatina et al., 2020 has not yet been published in the English version of the journal. We have referred to make it available to future readers of the journal. We have only the author's proofreading of this article in English. We send to the editor of the journal “Toxins” english и russion versions of both articles. Both articles do not contain the material described in the work submitted to the journal Toxins.

Reviewer 3 Report

I enjoyed reading the article „A Monoclonal Antibody against the C-Terminal Domain of Bacillus cereus Hemolysin II Inhibits HlyII Cytolytic Activity”. The study is well-written with a good readability. Research on Bacillus cereus are very important due to the ubiquitous presence of this pathogen and its strong role in foodborne illnesses.

The summary is consistent with the content of the work. The reviewed work presents the current state of knowledge in a logical, understandable, comprehensive, carefully and correctly citing literature - 28 items. It does not contain factual errors, the names are in accordance with generally accepted principles. Very nice figure number 5.

In my opinion, the text in English does not contain language errors.

Overall assessment of work very good.

My comments concern only the appearance of the figures 2. ­- please make it more explicit.

Author Response

Response to Reviewer #3

We are sincerely grateful for the appreciation of our work.

According to your remarks, we have changed the design of Figure 2 and the description in the text.

Reviewer 4 Report

the paper you propose “Monoclonal antibody against C –terminal domain of Bacillus cereus hemolysin II inhibits HlyII cytolytic activity” is overall well presented and it is focused on a topic quite relevant nowadays as it is the applications related to monoclonal antibodies.

However, some points should be better addressed.

In particular, the authors should complete the kinetic profile of the antibody candidates they present   corroborating their data with a SPR which would confirm the kinetic constants they have obtained from the indirect ELISA test.

Moreover, the Authors are suggested to perform SEC (size exclusion chromatography), in order to confirm the antibody’s profile and exclude the possible presence of aggregates which may be detrimental for their function and biological activity.

English language requires minor spelling revisions. Accept after revisions.

Author Response

The Authors are grateful to Reviewer #4 for detailed analysis and valuable comments and remarks.

We are very grateful to the reviewer for reminding us of an important and interesting method like SPR. Since in our study there is only one antibody, which is perfectly stained in ELISA, and for us, the relative interaction with toxins containing HlyIICTD, and the HlyIICTD itself, was of fundamental importance, we have not used this method. We suppose to use this method in future research.

Thanks for the important note, instead of SEC we centrifuged the antibody preparations to remove aggregates, the absence of aggregates was confirmed by the Dynamic light scattering measurement. Unfortunately, we missed to mention this fact in the original version of the manuscript, as this is a routine method used by us. Corrections made, section 2.1, lines 74-77, section 5.4, lines 347-351.